# Video Language Planning

**Yilun Du**[†‡], **Mengjiao Yang**[†§], **Pete Florence**[†], **Fei Xia**[†], **Ayzaan Wahid**[†], **Brian Ichter**[†], **Pierre Sermanet**[†], **Tianhe Yu**[†], **Pieter Abbeel**[§], **Joshua B. Tenenbaum**[‡], **Leslie Kaelbling**[‡] **Andy Zeng**[†], **Jonathan Tompson** [†]

Google Deepmind[†], Massachusetts Institute of Technology[‡], UC Berkeley[§]
https://video-language-planning.github.io/

## Abstract

We are interested in enabling visual planning for complex long-horizon tasks in the space of generated videos and language, leveraging recent advances in large generative models pretrained on Internet-scale data. To this end, we present *video language planning* (VLP), an algorithm using a tree search procedure, where we train (i) vision-language models to serve as both policies and value functions, and (ii) text-to-video models as dynamics models. VLP takes as input a long-horizon task instruction and current image observation, and outputs a long video plan that provides detailed multimodal (video and language) specifications that describe how to complete the final task. VLP scales with increasing computation budget where more computation time results in improved video plans, and is able to synthesize long-horizon video plans across different robotics domains – from multi-object rearrangement, to multi-camera bi-arm dexterous manipulation. Generated video plans can be translated into real robot actions via goal-conditioned policies, conditioned on each intermediate frame of the generated video. Experiments show that VLP substantially improves long-horizon task success rates compared to prior methods on both simulated and real robots (across 3 hardware platforms).

## 1 Introduction

Intelligently interacting with the physical world involves planning over both (i) high-level semantic abstractions of the task (i.e., what to do next), as well as the (ii) low-level underlying dynamics of the world (i.e., how the world works). Factorizing the planning problem into two parts, one driven by task-specific objectives and the other a task-agnostic modeling of state transitions, is an idea that is pervasive and fundamental. This factorization drives much of the classic work in robotics from integrating task and motion planning (Cambon et al., 2009; Wolfe et al., 2010; Kaelbling & Lozano-Pérez, 2011) to deriving control policies that can perform complex manipulation tasks over long time horizons such as tidying a dining table or rearranging a collection of objects to build new structures.

Pre-trained large language models (LLMs) (Brown et al., 2020; Chowdhery et al., 2022) have shown to be capable of generating high-level step-by-step plans for long-horizon tasks over symbolic (often linguistic) abstractions of the task (Huang et al., 2022a; Ahn et al., 2022), but this is only part of the solution. LLMs are restricted by what they can represent in text, and struggle with grounding i.e., reasoning over shapes, physics, and constraints of the real world (Tellex et al., 2020; Huang et al., 2023). LLMs can be integrated into larger vision-language models (VLMs) (Driess et al., 2023) that, when trained on sufficient data, can respect physical constraints observed in image inputs to generate more feasible plans that may be less likely to command the robot to perform impossible actions, or manipulate inaccessible objects. However, existing VLMs are predominantly trained on static image captioning and Q&A datasets – consequently, they continue to struggle to reason over dynamics e.g., how objects may move or collide with one another over time.

Meanwhile, recent text-to-video models trained on the wealth of videos on the Internet (Villegas et al., 2022; Ho et al., 2022), have demonstrated an ability to learn the dynamics and motions of objects by synthesizing video predictions of the future (Du et al., 2023b). Existing video models can only generate short time horizon clips without losing visual fidelity, and whether they can be applied for long-horizon planning remains unclear. Nevertheless, they exhibit properties that are complementary to VLMs in that they (i) can model the low-level visual dynamics of objects in ways that are more information-rich than text, and (ii) can absorb another source of Internet data e.g., YouTube videos. This leads to the natural question of how to build a planning algorithm that can

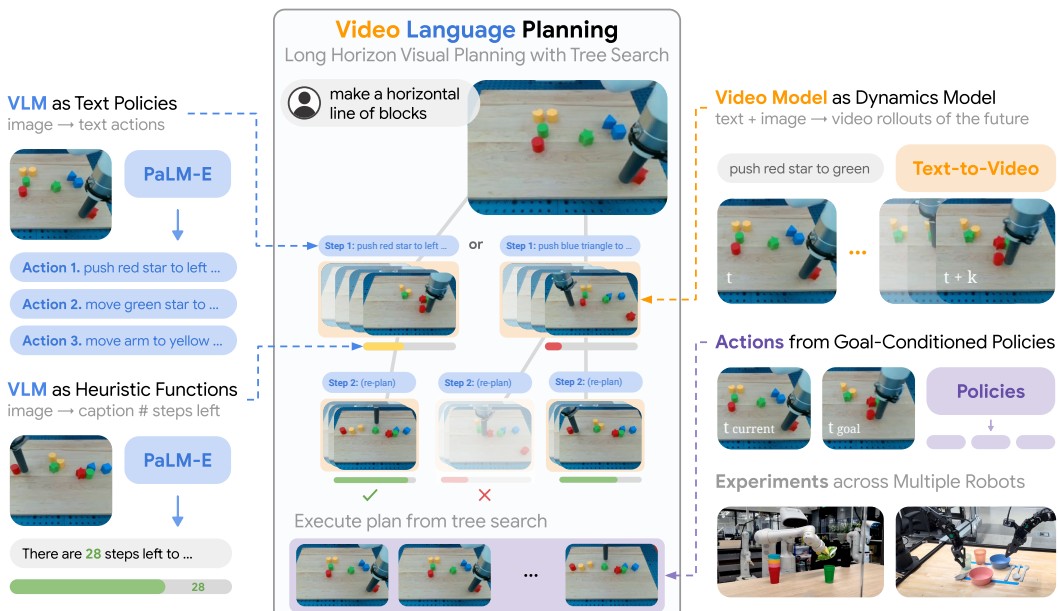

Figure 1: **Video Language Planning** uses forward tree search via vision-language models and text-to-video models to construct long-horizon video plans. From an image observation, the VLM policy (top left) generates next-step text actions, which a video model converts into possible future image sequences (top right). Future image states are evaluated using a VLM heuristic function (bottom left), and the best sequence is recursively expanded with tree search (middle). Video plans can be converted to action execution with goal-conditioned policies (bottom right).

leverage both long-horizon abstract planning from LLMs / VLMs and detailed dynamics and motions from text-to-video-models.

In this work, we propose to integrate vision-language models and text-to-video models to enable *video language planning* (VLP), where given the current image observation and a language instruction, the agent uses a VLM to infer high-level text actions, and a video model to predict the low-level outcomes of those actions. Specifically, VLP (illustrated in Fig. 1) synthesizes video plans for long-horizon tasks by iteratively: (i) prompting the VLM as a policy to generate multiple possible next-step text actions, (ii) using the video model as a dynamics model to simulate multiple possible video rollouts for each action, and (iii) using the VLM again but as a heuristic function to assess the favorability of each rollout in contributing task progress, then recursively re-planning with (i).

The combination of both models enables forward tree search over the space of possible video sequences to discover long-horizon plans (of hundreds of frames) that respect visual dynamics. In particular, VLP offers advantages in that it (i) can generate higher quality plans at inference time by expanding the branching factor of the search, allowing plan quality to scale with increasing compute budget, and (ii) benefits from training on incomplete language-labeled video data, which may contain short-horizon snippets (that can be re-composed and sequenced into long-horizon ones), or segments of videos with missing language labels (but contain dynamics that the video model can still learn from).

Experiments in both simulated and real settings (on 3 robot hardware platforms) show that VLP generates more complete and coherent multimodal plans than baselines, including end-to-end models trained directly to generate long videos conditioned on long-horizon task instructions. VLPs exhibit improved grounding in terms of the consistency of scene dynamics in video plans, and when used in conjunction with inverse dynamics models or goal-conditioned policies to infer control trajectories (Du et al., 2023b), can be deployed on robots to perform multi-step tasks – from picking and stowing a variety of objects over countertop settings, to pushing groups of blocks that rearrange them into new formations. In terms of execution success, VLP-based systems are far more likely to achieve task completion for long-horizon instructions than state-of-the-art alternatives, including PaLM-E (Driess et al., 2023) and RT-2 (Brohan et al., 2023) directly fine-tuned for long-horizon tasks. We also observe that when co-trained on Internet-scale data, VLP generalizes to new objects and configurations.

Our main contributions are: (i) video language planning, a scalable algorithm for generating long-horizon video plans by synergising vision-language models and text-to-video models, (ii) experiments

that show how VLP enables both simulated and real robots to perform complex long-horizon manipulation tasks, exhibiting task completion rates that often exceed that of the next best available approach by a significant margin, and (iii) ablations that study modes of generalization, as well as how VLP scales with more compute. VLP presents a modern re-examination of visual planning by integrating large generative models pretrained on Internet-scale data, and is not without limitations.

## 2 Video Language Planning

Our planning system, VLP, takes a visual observation $x_0$ of a scene and a natural language goal $g$ and infers a video plan $\{x_t\}_{1:T}$, where each image $x_t$ is as a sub-goal to accomplish $g$. We assume that each image $x_t$ serves as an accurate representation of world state and use an image goal-conditioned policy as a controller to infer low level control actions $u$ to reach each image state.

Below, we first discuss how vision-language and video models are used in VLP as planning sub-modules in Sec. 2.1. Next, we talk about our tree-search algorithm using vision-language and video building blocks in Sec. 2.2. Finally, in Sec. 2.3, we discuss how we can convert video plans into policies to accomplish each long-horizon task.

### 2.1 Using Vision-Language and Video Models as Planning Submodules

We first discuss how to use vision-language and video models as sub-modules in VLP to synthesize long horizon plans. At a high level, we use the multimodal processing power of VLMs to propose abstract text actions $a^i$ to execute given goals and images. We then use the dynamics knowledge of video models to accurately synthesize possible future world states $x_{1:T}^i$ when abstract actions are executed. Finally, we use a VLM to process possible future world states $x_{1:T}^i$ and assess which sequence $x_{1:T}$ and associated actions are the most promising to complete a task.

**Vision-Language Models as Policies.** Given a high-level goal $g$, VLP searches over a space of possible abstract actions; these text actions $a$ are generated by a VLM policy $\pi_{\text{VLM}}(x, g) \rightarrow a$ that is conditioned both on the goal $g$ and an image $x$ of the current state. We implement this policy following Driess et al. (2023) and query the VLM for a natural language action to take given as context the natural language goal and a tokenized embedding of the current image (Fig. 1 Top Left). We experiment with two different strategies for constructing this policy. In the first, we provide the VLM a set of example text action labels and ask the VLM to predict possible actions to accomplish a goal. In the second, we finetune the PaLM-E model on randomly selected short trajectory snippets $x_{1:S}$ labeled with abstract actions inside a long trajectory $x_{1:H}$ that accomplishes a long horizon goal $g$.

**Video Models as Dynamics Models.** In order to perform the high-level search, given an image $x$ of a current state and a language description of an abstract action, we need to predict the concrete resulting state. In addition, to generate low-level controls that instantiate this abstract action, we need a feasible sequence of low-level states that "interpolate" between the current state and the resulting state. We obtain both of these things from a text-to-video model $f_{\text{VM}}(x, a)$, which takes an image $x$ and a short horizon text instruction $a$ and outputs a short synthesized video $x_{1:S}$ starting at the image observation $x_0$ (Fig. 1 Top Right) following Du et al. (2023b). We construct this text-to-video model by training on a set of short image trajectory snippets $x_{1:T}$ and associated language labels $a$.

**Vision-Language Models as Heuristic Functions.** To effectively prune branches in search, we use a VLM to implement a heuristic function $H_{\text{VLM}}(x, g)$ which takes as input an image observation $x$ and a natural language goal description $g$ and outputs a scalar "heuristic" predicting the number of actions required to reach a state satisfying goal $g$ from current state $x$ (Fig. 1 Bottom Left). To construct this heuristic function, we finetune a PaLM-E model using long trajectory snippets $x_{1:H}$ which accomplish a long horizon goal $g$, and train it to predict, given an image in the subtrajectory $x_t$, the number of steps left until the end of the trajectory snippet. The negated number of predicted steps to goal completion from VLM is used to implement $H_{\text{VLM}}(x, g)$ (so that high heuristic value corresponds to being close to goal completion).

### 2.2 Planning with Vision-Language Models and Video Models

Given a combination of modules discussed in Sec. 2.1, directly applying the $\pi_{\text{VLM}}$ to infer text actions $a$ to reach goal $g$ is not sufficient, as $\pi_{\text{VLM}}$ is not able to perform sufficiently accurate long-horizon reasoning to select actions that are helpful in the long run. Furthermore, there are many possible low-level image sub-sequences that correspond to different ways to perform $a$, but it is critical to select one that is consistent with the rest of the actions that must be taken to reach $g$.

---

**Algorithm 1** Decision Making with VLP

1: **Input:** Current visual observation $x_0$, Language goal $g$
2: **Functions:** VLM Policy $\pi_{\text{VLM}}(x, g)$, Video Model $f_{\text{VM}}(x, a)$, VLM Heuristic Function $H_{\text{VLM}}(x, g)$
3: **Hyperparameters:** Text-Branching factor $A$, Video-Branching factor $D$, Planning Beams $B$, Planning horizon $H$
4: plans $\leftarrow [[x_0] \ \forall \ i \in \{1 \dots B\}]$        # Initialize B Different Plan Beams
5: **for** $h = 1 \dots H$ **do**
6: **for** $b = 1 \dots B$ **do**
7:  $x \leftarrow$ plans$[b][-1]$       # Get the Latest Image State in the Plan Beam
8:  $a_{1:A} \leftarrow \pi(x, g)$         # Generate $A$ Different Text Actions
9:  video_branches $\leftarrow [f_{\text{VM}}(x, a_i)$ for i in $(1 \dots A)$ for j in $(1 \dots D)]$
10:  plans$[b]$.append(argmax(video_branches, $H_{\text{VLM}}$))  # Add Video with Highest Value to Plan
11: **end for**
12: max_idx, min_idx $\leftarrow$ argmax(plans, $H_{\text{VLM}}$), argmin(plans, $H_{\text{VLM}}$)
13: plans$[$min_idx$] \leftarrow$ plans$[$max_idx$]$   # Periodically Replace the Lowest Value Plan
14: **end for**
15: plan $\leftarrow$ argmax(plans, $H_{\text{VLM}}$)        # Return Highest Value Plan

---

Instead, we propose to search for a sequence of actions to reach $g$, corresponding to finding a long-horizon video plan $x_{1:H}$ which optimizes

$$x^*_{1:H} = \underset{x_{1:H} \sim f_{\text{VM}}, \pi_{\text{VLM}}}{\arg\max} \ H_{\text{VLM}}(x_H, g). \tag{1}$$

A separate procedure is then used to instantiate control actions $u$ to enact the optimized video plan $x^*_{1:H}$. To sample long-horizon video plans $x_{1:H}$, we first synthesize a short horizon video plan $x_{1:S}$ from a starting image $x$ through $x_{1:S} = f_{\text{VM}}(x, \pi_{\text{VLM}}(x, g))$ and autoregressively extend to a full long-horizon video plan by recursively applying $f_{\text{VM}}(x, \pi_{\text{VLM}}(x, g))$ on the final synthesized image state $x_S$. To optimize across video plans in Eqn (1), we use a tree-search procedure based on parallel hill climbing (Selman & Gomes, 2006) (illustrated in Algorithm 1).

Our planning algorithm initializes a set of $B$ parallel video plan beams. At each step of the planning horizon, for each video beam, we first sample a set of $A$ actions using $\pi_{\text{VLM}}(x, g)$, and for each action we synthesize $D$ different videos using $f_{\text{VM}}(x, a)$. We then use our heuristic function $H_{\text{VLM}}(x, g)$ to select the generated video with the highest heuristic among the $A \times D$ generated videos and extend the corresponding video plan beam with this generated video. Over the course of plan generation, certain video plan beams will obtain high heuristic value and be more promising to explore. Therefore, every 5 steps, we discard the beam with the lowest value and replicate its video plan with the beam with the highest value. Our final full long horizon video plan corresponds to the beam with highest heuristic value at the end of planning.

**Preventing Exploitative Model Dynamics.** When our planning procedure optimizes the VLM heuristic function $H_{\text{VLM}}(x, g)$ it can exploit irregularities in the dynamics model $f_{\text{VM}}(x, a)$ to get artificially high estimates. For instance, the planning procedure can exploit videos from $f_{\text{VM}}(x, a)$ where key objects have teleported to desired locations or where the final image observation obscures undesirable portions of world state. To prevent over-exploitation of $H_{\text{VLM}}(x, g)$, during the planning procedure in Algorithm 1, we discard generated videos from $f_{\text{VM}}(x, a)$ if they increase the the heuristic estimate $H_{\text{VLM}}(x, g)$ above a fixed threshold.

## 2.3 ACTION REGRESSION FROM VIDEO THROUGH GOAL-CONDITIONED POLICIES

Given a synthesized video plan $x_{1:H}$, to execute tasks, we must infer control actions $u$ to reach synthesized images. Prior work infers actions from synthesized videos by using an inverse-dynamics model on each synthesized frame (Du et al., 2023b).

In many settings, a single action may not be sufficient to directly reach the next synthesized image, *i.e.* if you need to remove the cap off a toothbrush, and even in settings in which this is the case, it may be difficult to precisely predict the correct action to reach the next frame. To reduce the burden on the inverse dynamics model, we propose to use a short-horizon goal-conditioned policy $\pi_{\text{control}}(x, x_g)$, which given the current image observation $x$ and next frame in a video plan $x_g$ outputs a low level control action $u$ that makes progress towards $x_g$. For each frame in our video plan $x_{1:H}$, the goal-conditioned policy is executed for a fixed pre-specified number of timesteps. We train $\pi_{\text{control}}$ using paired image and low level control snippets $x^i_{1:T}$ and $u^i_{1:T}$, where we sample a random timestep $t$, a corresponding state $x_t$, and future state $x_{t+h}$, and train $\pi_{\text{control}}(x_t, x_{t+h})$ to predict $u_t$.

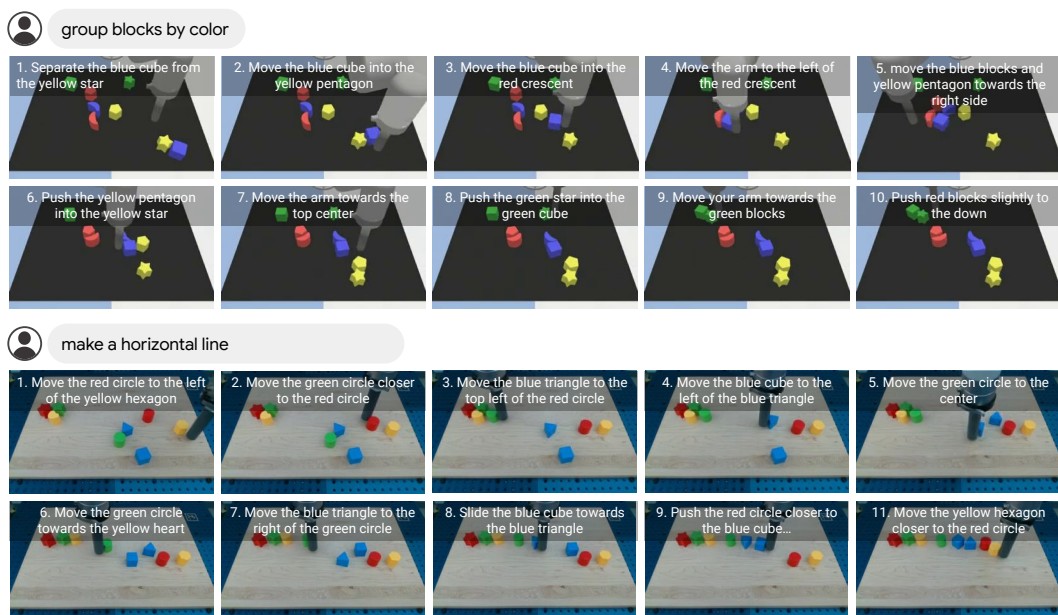

Figure 2: **Long Horizon Video Plan.** Long horizon video plans generated by VLP on both simulated and real images. VLP is *only given* the *initial image* and *language goal*. Language subplans and other image frames are *directly synthesized*.

|  | Sim Environment | | | Real Environment | | |
|---|---|---|---|---|---|---|
| Model | Move Area | Group Color | Make Line | Move Area | Group Color | Make Line |
| UniPi | 2% | 4% | 2% | 4% | 12% | 4% |
| VLP (No Value Function) | 10% | 42 % | 8% | 20% | 64% | 4% |
| VLP (Ours) | **58%** | **98%** | **66%** | **78%** | **100%** | **56%** |

Table 1: **Accuracy of Generated Video Plans.** The percentage VLP and baselines are able to synthesize a full video plan which can fully complete tasks in simulation and real environments. VLP substantially outperforms both UniPi and directly combining the VLM policy

**Replanning.** Given a very long-horizon task, it is both difficult to use $\pi_{\text{control}}$ to accurately execute the full video plan $x_{1:H}$ (due to accumulating error) and difficult to fully synthesize a plan that completely finishes a long-horizon task given a fixed planning horizon. To circumvent this issue, we use receding horizon control strategy (Kwon & Han, 2005), where we generate videos plans with a fixed horizon (that might not fully complete the task), and then repeatedly regenerate/replan video plans with the same horizon after a fixed number of action executions.

## 3 EXPERIMENTAL RESULTS

We first evaluate the ability of VLP to synthesize long-horizon video plans for different tasks in Sec. 3.1. We then investigate VLP's ability to execute generated video plans in various environments in Sec. 3.2. Finally, we further investigate generalization capabilities of VLP in Sec. 3.3.

### 3.1 LONG-HORIZON VIDEO SYNTHESIS

**Baselines.** We compare our approach with two other approaches for synthesizing long-horizon video plans. First, we consider training a text-to-video model $f_{\text{VM}}$ on long horizon text goals, as in UniPi (Du et al., 2023b), omitting the entire VLP planning process. Next, we consider synthesizing long horizon video plans by chaining $\pi_{\text{VLM}}$ policy with $f_{\text{VM}}$, without the heuristic function.

**Object Rearrangement.** We first illustrate video plans in the Language Table environment (Lynch et al., 2023), which consists of a dataset of long-horizon demonstrations and text labels with incomplete short horizon labels (Appendix A.3). We give as input to VLP a random image and randomly chosen language goal. We then visualize the generated VLP plans (Fig. 2). We report the quantitative success of synthesizing long-horizon videos given random starting images for each task in Language Table in Tab. 1. For each reported number, we generated a total of 50 videos from each method and visually assessed the percentage of time the video successfully solved the given task. VLP substantially outperforms the baseline of directly synthesizing videos given a long-horizon prompt,

| Beams | Language Branch | Video Branch | Line Performance |
|:---:|:---:|:---:|:---:|
| 1 | 1 | 1 | 4% |
| 1 | 1 | 4 | 10% |
| 1 | 4 | 4 | 22% |
| 2 | 4 | 4 | **56%** |

Beam 1, Branch 1

Beam 2, Branch 16

Figure 3: **Video Accuracy vs Planning Budget.** *Left*: VLP scales positively with more compute budget; it is better able to synthesize plans to solve tasks with more planning (i.e. with a higher beam-search branching factor). Success percentage reported on the make line task. *Right*: Qualitative illustration of video plans for making a line generated without planning (Beam 1, Branch 1) compared to extensive planning (Beam 2, Branch 16).

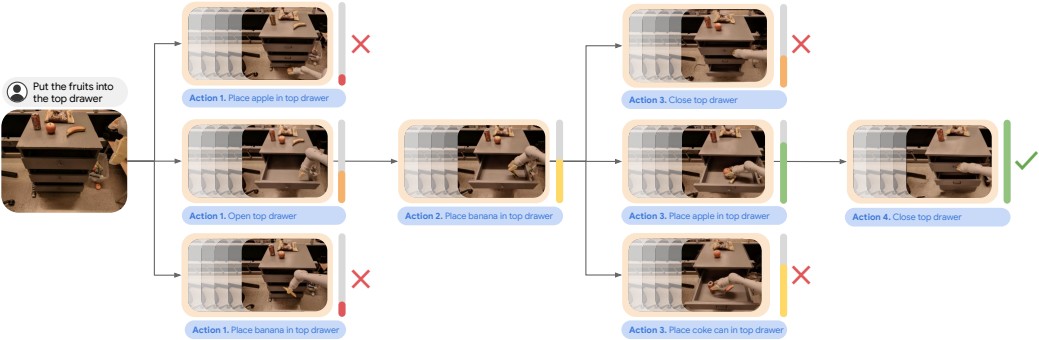

Figure 4: **Planning Tree on 7DoF Mobile Manipulator.** VLP is able to prune unlikely language and video branches to synthesize a coherent long-horizon video plan.

indicating the importance of hierarchical structure. VLP further outperforms the ablation of only using a VLM policy with a video model, pointing to the effectiveness of the VLP planning procedure and including the value function.

**Effect of Search of Video Synthesis.** We analyze the effect of search in generating long-horizon videos in Fig. 3 (left). We consider increasing the video branching, language branching and the beams in the search procedure. We find that each increase of branching factor in search substantially increases the success of synthesized long horizon plans. A qualitative illustration of the difference of generated plans with small and large branching factor is illustrated in Fig. 3 (right).

**Planning on 7DoF Mobile Manipulators.** We qualitatively illustrate how we can generate plans on a higher-DoF, 7DoF Mobile Manipulator in Fig. 4. Our planning system is able to generate videos of actions that both open and close drawers in order to satisfy specified text prompts.

**Planning on Multicamera 14DoF Bi-Manual Manipulators.** We further illustrate how our approach can generate multi-view 4-camera videos of dexterous manipulation on the 14DoF bi-manual ALOHA (Zhao et al., 2023) platform in Fig. 5. Our video model outputs videos across views simultaneously (by concatenating each view channelwise), while our VLM policy and heuristic function takes as input top and side views. While short horizon text labels are incomplete (Appendix A.3), our approach synthesizes multiview consistent plans which stack bowls, cups, and utensils.

### 3.2 LONG-HORIZON EXECUTION

We next evaluate the ability of VLP to not only generate plans as in Sec. 3.1, but to actually use planning (and replanning) to *execute* long-horizon tasks in closed-loop environments.

**Baselines.** We compare our approach to a set of approaches to solve long-horizon tasks. (i) We consider using a VLM to directly plan, using PaLM-E (Driess et al., 2023) to plan short horizon text snippets to execute, which are converted to actions using a text-conditioned policy, conditioned on generated text snippets from PaLM-E. We also (ii) compare with UniPi (Du et al., 2023b), where videos are directly generated by a text-to-video model trained on long-horizon text goals and converted to actions using our goal-conditioned policy. Next, we consider (iii) directly learning language-conditioned behavioral cloning policy on long-horizon text and actions, using the codebase and architecture of the LAVA model Lynch et al. (2023). Finally, we (iv) compare with a multi-modal transformer architecture to predict actions, and finetune the RT2 model Brohan et al. (2023) on long-horizon text and actions. Note that methods that plan in language space (Hao et al., 2023) are not applicable, as they cannot simulate detailed visual dynamics (Appendix A.1).

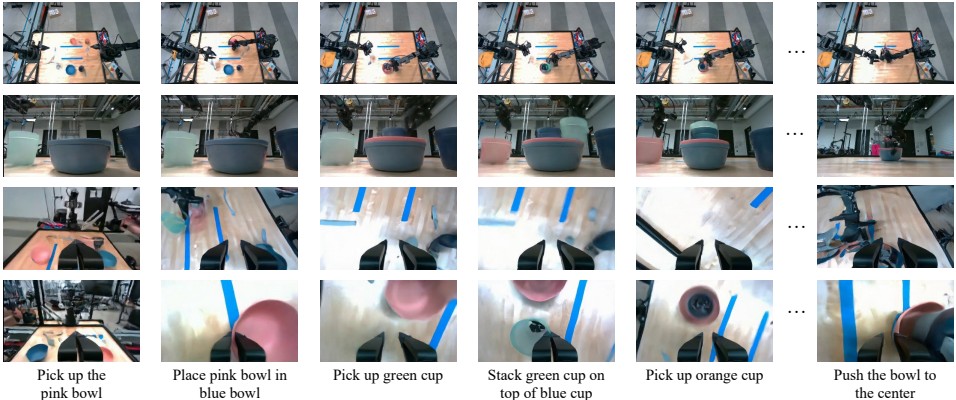

| | | | | | |
|---|---|---|---|---|---|
| Pick up the pink bowl | Place pink bowl in blue bowl | Pick up green cup | Stack green cup on top of blue cup | Pick up orange cup | Push the bowl to the center |

Figure 5: **Multiview Video Plans for Dexterous Manipulation.** Long horizon video plans (and associated language subgoals) generated by VLP for solving the long horizon task of stacking everything in a table together. VLP is able to synthesize multiview video plans across 4 cameras, that are consistent with each other and with task completion. The first 5 generated language subgoals goals are illustrated as well as the final generated goal image. VLP is only given first image.

| Model | Move to Area | | Group by Color | | Make Line | |
|---|---|---|---|---|---|---|
| | Reward | Completion | Reward | Completion | Reward | Completion |
| UniPi (Du et al., 2023b) | 30.8 | 0% | 44.0 | 4% | 44.0 | 4% |
| LAVA (Lynch et al., 2023) | 59.8 | 22% | 50.0 | 2% | 33.5 | 0% |
| RT-2 (Brohan et al., 2023) | 18.5 | 0% | 46.0 | 26% | 36.5 | 2% |
| PALM-E (Driess et al., 2023) | 36.5 | 0% | 43.5 | 2% | 26.2 | 0% |
| VLP (Ours) | **87.3** | **64%** | **95.8** | **92%** | **65.0** | **16%** |

Table 2: **Execution Performance on Long Horizon Tasks.** VLP is able to accurately execute actions for different long-horizon synthetic language table tasks. VLP substantially outperforms all existing methods.

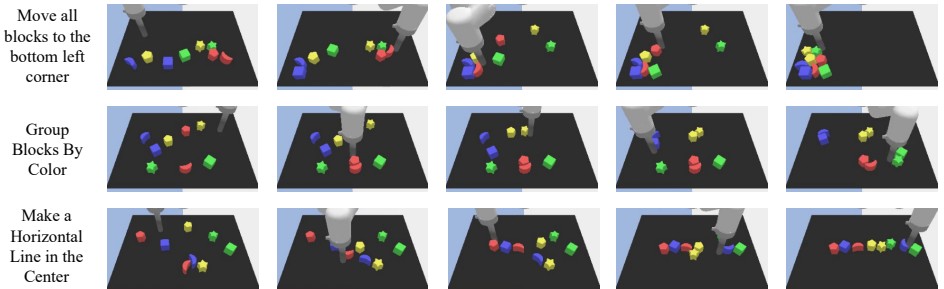

Figure 6: **Simulation Execution.** Illustration of execution of VLP on different simulated environments. VLP is able to accomplish different long horizon goals.

**Quantitative Results.** We evaluate each approach quantitatively on moving all blocks to different areas of the board, grouping blocks by color, or making blocks in a line (details on quantitative evaluation in Appendix). We report quantitative results in Tab. 2, and find that our approach substantially outperforms all baseline methods. As the task horizon of each task is very long (around 1500 steps), we found that many baseline methods would become "stuck" and stop acting effectively. We illustrate example executions using VLP in Fig. 6.

**Effect of Planning.** Next, we analyze the effect of the amount of planning on execution success rates in Tab. 3. We find that increasing both the planning horizon and the branching factor of planning substantially improves the success of task execution (at the cost of inference time).

**Ablations of Goal-Conditioned Policy** We further conduct experiments on different approaches to extracting actions from videos in Tab. 4. We find that using a goal-conditioned policy conditioned on each intermediate frame in a synthesized video leads to the best overall performance (outperforming using a goal-conditioned policy sparsely on the end frames of each short-horizon video).

**Real Execution.** We provide executions of VLP on multiple real-world robots in Fig. 7 and Fig. 8. As in Fig. 7, VLP is able to effectively execute each shown long-horizon task on a real Language Table robot. We further provide executions of generated video plans of our approach on the 7DoF mobile manipulator in Fig. 8. Similarly we find that a goal-conditioned policy can realize plans.

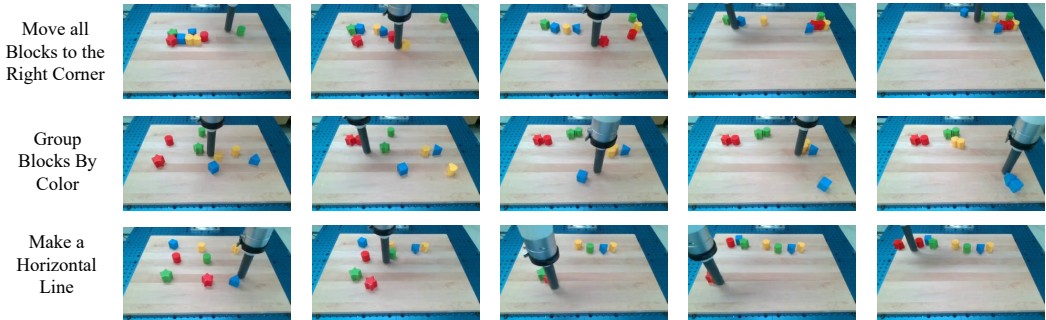

Figure 7: **Real Execution.** Illustration of execution VLP on real world robots. VLP is able to accomplish different long horizon goals when executed on real robots.

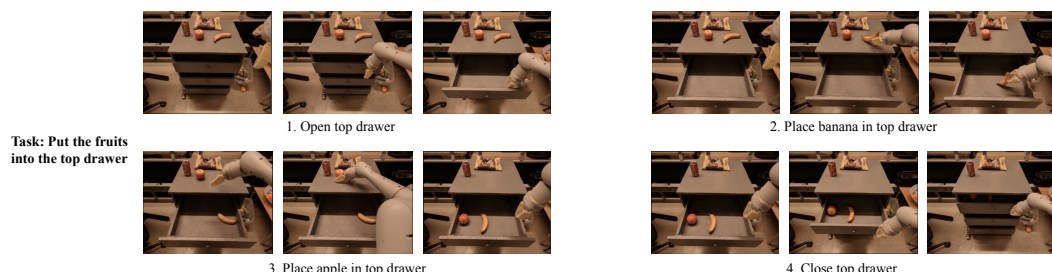

Figure 8: **7DoF Mobile Robot Execution.** VLP is able to execute complex, long horizon plans on mobile robot.

| Beams | Planning Horizon | Branching Factor | Line Score | Line Completion |
|---|---|---|---|---|
| 1 | 1 | 4 | 48.9 | 0% |
| 1 | 1 | 16 | 53.3 | 2% |
| 1 | 2 | 16 | 58.1 | 8% |
| 2 | 2 | 16 | **65.0** | 16% |

Table 3: **Execution Accuracy vs Planning Budget.** VLP is able to more accurately execute video plans to solve tasks with a larger amount of planning. Success percentage reported on the make line task.

| Action Inference | Group Color Score | Group Color Completion |
|---|---|---|
| Inverse Dynamics | 89.7 | 80% |
| Goal Policy (Last) | 85.0 | 66% |
| Goal Policy (Every) | **95.8** | **92%** |

Table 4: **Extracting Actions From VLP Video Plans.** Comparison of using inverse dynamics or applying a goal-conditioned policy to either the last frame or every frame of synthesized short-horizon video. Success percentage reported on the group by color task.

## 3.3 GENERALIZATION

**Generalization to Lighting and Objects.** In VLP, policy execution is abstracted into visual goal generation followed by a goal-conditioned controller. With this abstraction, a video model can simply focus on capturing the visual dynamics objects, while a goal-conditioned policy needs to focus only on relevant visual details to achieve the next (nearby) goal. We found that this enables VLP to generalize well, as the video model is able to visually generalize to new images, while the policy is able to generalize well to nearby new visual goals. In Fig. 9 (top), VLP is able to generalize the task of putting all objects in top right corner, to three new objects, a rubber donut and cupcake and a wooden hexagon. In Fig. 9 (bottom) VLP is further able to generalize to lighting conditions substantially different than the ones the model was trained on and can be successfully deployed to a new robot in a different building location.

**Generalization to New Tasks** In VLP, both VLM and text-to-video models may be pre-trained on a vast amount of Internet data. In Fig. 10, we train both VLM and text-to-video models on a large mix of datasets and illustrate how it further generalizes and executes new tasks on unseen objects.

## 4 RELATED WORK

There is a great deal of recent work in leveraging large pretrained foundation models for decision-making agents (Yang et al., 2023b) – in particular, using LLMs (and the commonsense knowledge they store) to infer actions represented in text. For example, given language instructions, LLMs can be prompted to generate high-level step-by-step plans (Huang et al., 2022a; Ahn et al., 2022; Wang et al., 2023a) – each step mapped robot actions, invoked with pre-trained policies or existing APIs (Liang et al., 2023). Using existing VLMs (Chen et al., 2022), plans can also be conditioned on visual inputs, represented with language descriptions (Zeng et al., 2022; Huang et al., 2022b) or token embeddings co-trained with LLMs (Driess et al., 2023). Nevertheless, VLMs are often

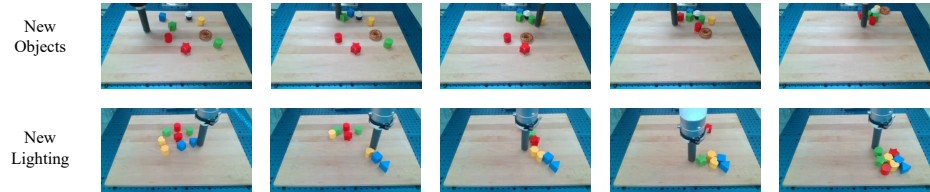

Figure 9: **Generalization to Objects and Lighting.** VLP can generalize execution to scenes with new objects (top) consisting of a wooden yellow hexagon, rubber donut and rubber cupcake and to a new office (bottom) with different lighting conditions (a lot more lighting on the right side of the board) and similarly execute tasks.

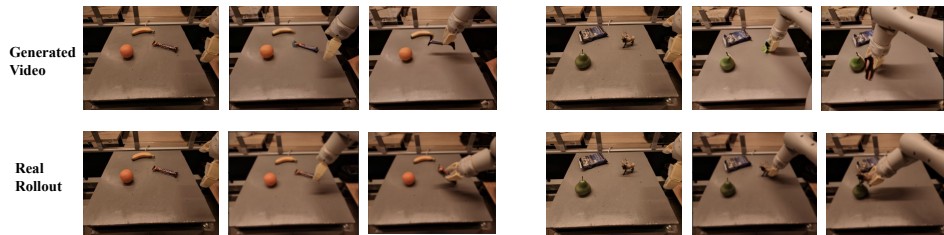

*Novel instruction: Pick snicker energy bar*          *Novel instruction: Move moose toy near green pear*

Figure 10: **Task Generalization** VLP can generalize to new tasks on unseen objects using internet knowledge.

pretrained on static image datasets (i.e., image-in, text-out), and subsequently (i) are limited to plans that can be expressed in text, and (ii) may struggle to reason about the dynamics of the world.

Video models, on the other hand, exhibit potential to generate informative image sequences of the future, and there exists a body of work using them as dynamics models to capture object motion across image states (Finn et al., 2016; Xue et al., 2016; Oprea et al., 2020; Oh et al., 2015). Generating high-quality videos over long time horizons is a known challenge (Babaeizadeh et al., 2021; Saxena et al., 2021); however, recent progress in larger models and text-conditioning give rise to a new generation of text-to-video models that can generate detailed image frames (Ho et al., 2022; Villegas et al., 2022), that can drive policies for decision making (Du et al., 2023b). However, these works focus predominantly on short-horizon videos. Our work investigates using text-to-video models together with VLMs to produce long-horizon video plans (potentially predicting hundreds of frames into the future), leveraging the scalability of tree search to plan in the space of videos and language.

Our work can be viewed as bringing together two families of generative foundation models to compose new capabilities – a strategy shown to be effective for applications such as 2D image synthesis (Du et al., 2020; Liu et al., 2021; Nie et al., 2021; Liu et al., 2022; Wu et al., 2022; Du et al., 2023a; Wang et al., 2023b), 3D synthesis (Po & Wetzstein, 2023), video synthesis (Yang et al., 2023a), trajectory planning (Du et al., 2019; Urain et al., 2021; Gkanatsios et al., 2023; Yang et al., 2023c) and multimodal perception (Li et al., 2022; Zeng et al., 2022). Most similar to our work, HiP (Ajay et al., 2023) combines language models, video models, and action models for hierarchical planning. In contrast, our work uses a forward search procedure to combine the strengths of a VLM and a video model. This composition through forward search enables us to simulate and reason about long horizons of future actions, while HiP only generates and acts on plans one step at a time.

## 5 LIMITATIONS AND CONCLUSION

**Limitations.** Our planning approach leverages images as a world state representation. In many tasks, this is insufficient as it does not capture the full 3D state and cannot encode latent factors such as physics or mass. Limitations can be partly remedied by generating multi-view videos or by using heuristic function with the full video plan as input. In addition, we observed that our video dynamics model does not always simulate dynamics accurately. In several situations, we observed that synthesized videos would make objects spontaneously appear, change shape, or teleport to new locations. We believe that larger video models or explicit reinforcement learning feedback for physics (Black et al., 2023) could help solve these problems. Finally, our video generation is slow, and it takes 30 minutes to synthesize a full long-horizon video plan, which can be accelerated by distillation (Salimans & Ho, 2022) or alternative video generation procedures (Villegas et al., 2022).

**Conclusion.** We have presented VLP, an approach to long-horizon decision-making by combining VLMs with text-to-video models. We illustrate the power of test-time composition, in this case planning, to generate behaviors much more complex than its components. While our experiments largely explore VLP in the context of robotics, there may be downstream applications in other areas too, such as steerable media generation as well as additional methods to compose models.

**Acknowledgements.** We would like to thank Tomas Lozano-Perez for providing helpful comments in the project and Kamyar Ghasemipour for help in setting up experiments on the Language Table real robot. We gratefully acknowledge support from NSF grant 2214177; from AFOSR grant FA9550-22-1-0249; from ONR MURI grant N00014-22-1-2740; from ARO grant W911NF-23-1-0034; from the MIT-IBM Watson Lab; from the MIT Quest for Intelligence; and from the Boston Dynamics Artificial Intelligence Institute. Yilun Du is supported by a NSF Graduate Fellowship.

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

# A  APPENDIX

In the Appendix, we provide additional results in Sec. A.1. We provide evaluation details in Sec. A.2, dataset details in Sec. A.3, training details in Sec. A.4, and planning details in Sec. A.5.

## A.1  ADDITIONAL RESULTS

Below, we provide additional experimental results.

**Failure Cases.**     We observed several failures cases when applying our approach across tasks. First, when using world knowledge from other text-to-video datasets to generalize and execute new tasks, we found that sometimes the video model would incorrectly interpret tasks as illustrated in Fig. XI, where it interprets the manipulator as an octopus. In addition, we found that our approach sometimes synthesized videos with inconsistent physics, where objects would infrequently appear or disappear as illustrated in Fig. XII.

**Robustness of Goal-Conditioned Policy**     We found that our decomposition of execution in VLP to visual goal generation and subsequent execution allowed for strong generalization in the goal-conditioned policy. This was because the goal-conditioned policy can discard most information in image goals and focus only on the portion required for immediate execution of the policy. We illustrate this robustness in Fig. XIII. Even when the generated goals are misaligned with the observed board, with many artifacts in the generated boundaries, our goal-conditioned policy is still able to execute the visual task.

**Additional Video Plans.**     We provide additional video plans synthesized by our approach on block arrangement in Fig. XIV. Our approach is able to robustly synthesize both language and video plans.

**Planning with Language Models.**     To directly plan with language models, we would need a language model to accurately simulate the visual dynamics of an image serialized to text. In Fig. XV, we find that existing LLMs are unable to accurately simulate visual dynamics, missing object collisions, identities and gripper movement, preventing them from being effectively used to plan.

## A.2  EVALUATION DETAILS

**Video Evaluation.**     To evaluate whether a generated video satisfied a long horizon task, we manually assessed whether at any point in the generated video, the image state in the video satisfied the specified long-horizon goal. Due to the computational cost of generating long-horizon video (taking approximately 30 minutes per video), we generated a total of 50 videos for each goal across each task and method.

**Execution Evaluation.**     We used the ground truth simulation state of each block in the Language Table environment to compute rewards and task completion thresholds for each task in the paper. To compute the reward function for "group blocks by color", we assessed the percentage of blocks of the same color that were within 0.1 units of each other. To compute the reward for "move blocks to an area", we assessed the number of blocks within 0.2 x units and 0.27 y units to a specified corner. To compute the reward for "move blocks into a line in the center of a board", we computed the number of blocks with 0.05 units of the center of the board. We gave all baselines and VLP a total of 1500 timesteps of execution in each environment, with performance measured at the end of 1500 timesteps. If at an intermediate timestep the task was complete, we terminated execution early.

We evaluated each method on a total of 50 environments on each task due to slowness in execution of both our method and baselines (VLP took approximately 1 hour per environment, while RT2 baselines took approximate 0.5 hours per environment). For VLP, we generated a plan of horizon length 2 with beam width 2 and branching factor 16, where each video in a planning horizon corresponded to 16 frames. We called the goal-conditioned policy a total of 4 times for each of the first 16 frames of the synthesized video plan in simulated environments. On real world evaluation tasks, we called the goal-conditioned policy on the first 10 frames of the synthesized video plan to more finely account for differences in physics of the real world execution compared to those used to gather the data.

## A.3  DATASET DETAILS

**Language Table.**     We trained VLP on approximately 10000 long horizon trajectories in both simulation and real across a set of several hundred different long horizon goals. We chose a total of 3 goals to run experiments on as they allowed easy automated evaluation. Data in simulation and real combined in total is roughly 20000 trajectories and with approximately 400000 short-horizon text labels. Since long-horizon goal demonstrations are gathered all at once, without any intermediate short-horizon text goals, short-horizon text is labeled in hindsight on long-horizon demonstrations. A

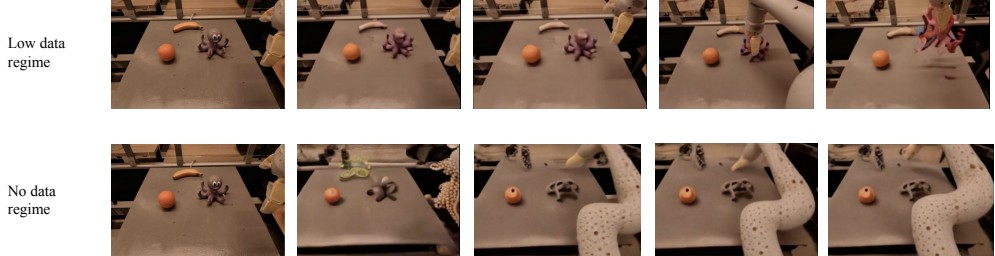

Figure XI: **Failure in Transferring Web Knowledge.** VLP is instructed to "pick up octopus toy". In the low-data regime, the model has only seen the octopus toy in training data a few times, and the generated frames roughly reconstructed the shape of the toy, but fails at accurately recover the dynamics. The the no-data regmie, the model has never seen in training data, and without any training data VLP transfers the wrong web knowledge and makes the gripper octopus arms.

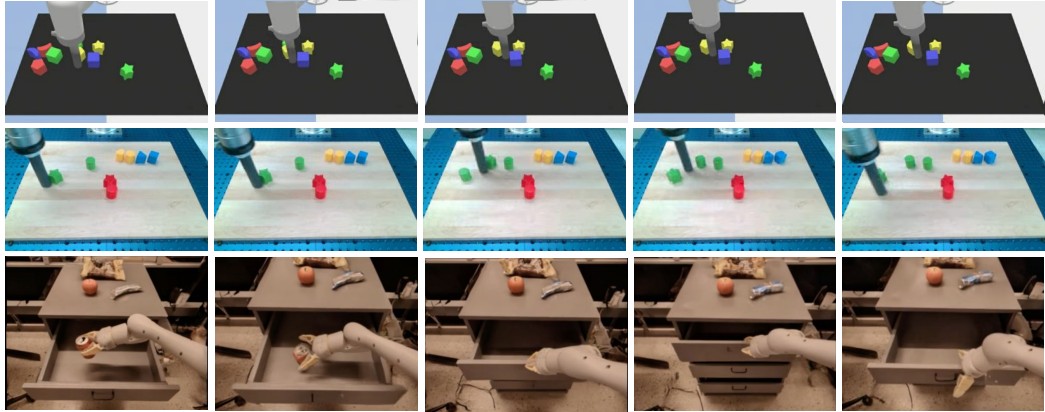

Figure XII: **Failure in Physics.** When synthesizing videos, VLP will sometimes make objects dissappear or reappear. Creating long videos that respect object permanence remains an important future direction for our work.

set of human raters are asked to go through long-horizon demonstrations and select sub-snippets that correspond to interpretable short-horizon text goals. As many robot motions do not easily map to text actions, these sub-snippets do not capture all frames in demonstrations and are often overlapping in time, leading to short-horizon text labels to be incomplete.

**7DoF Mobile Manipulator.** For our 7DoF mobile manipulator planning and execution experiments, we trained VLP on the dataset from RT-1 (Brohan et al., 2022). For our generalization experiments of VLP, we train a large text-to-video diffusion model using a mix of data from our particular 7DoF Mobile Manipulator, Bridge (Ebert et al., 2021), RT-2 (Brohan et al., 2023), Ego4D (Grauman et al., 2022), EPIC-KITCHEN (Damen et al., 2018), and LAION-400M (Schuhmann et al., 2022).

**14DoF Bi-Manual Manipulation.** For our 14DoF Bi-Manual Manipulation, we train VLP on approximately 1200 teleoped demonstrations on a kitchen stacking task, where operators were first asked to stack bowls on top of each other, then cups on top of bowls, and then utensils on top of cups. Each demonstration was annotated with roughly 20 language instructions, leading to roughly 25k short-horizon text labels. Similar to the Language Table setting, short-horizon text labels are incomplete and generated through hindsight relabeling of the long teleoped trajectories.

## A.4 TRAINING DETAILS

**Video Models.** To train video diffusion models, we follow the model architecture and training approach from (Du et al., 2023b). We train a base text-conditioned video generation model at $24 \times 40$ resolution and upsample it to $48 \times 80$ resolution and then $192 \times 320$ resolution, where videos at each resolution are set to generate a temporal video of length 16 frames. We use a base channel width of 256 across models and train a base text-conditioned video model using 64 TPUv3 pods for 3 days and higher resolution superresolution models for 1 day. We train separate text-to-video models per domain.

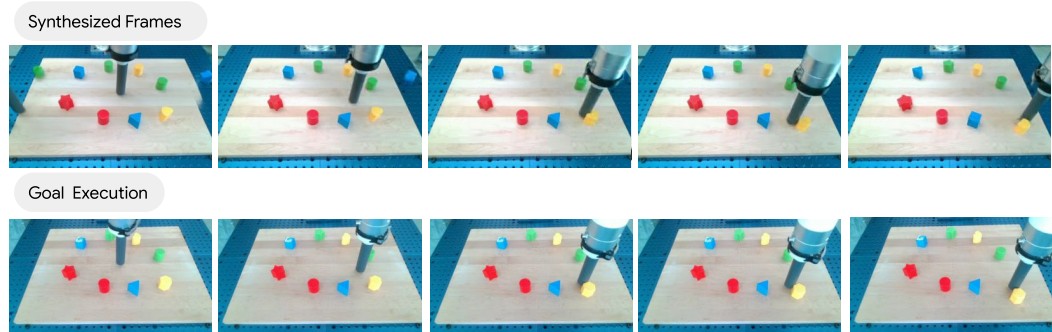

Figure XIII: **Goal Policy Robustness to Synthesized Goals** The goal-conditioned policy is robust to noise in synthesized goals. In the above example, given synthesized goals in the top row, the policy can directly execute a control actions in the real environment in the bottom row.

**VLM Models.** To train VLMs, we follow the architecture and codebase of PaLM-E (Driess et al., 2023). We fine-tune a single 12B PaLM-E (Driess et al., 2023) jointly to both predict heuristics values and policies. We formulate heuristic value prediction as a sequence prediction problem, allowing the heuristic functions to predict multiple different values for multi-modal demonstrations. We finetune the VLM model using 64 TPUv3 pods for 1 day on data in each domain and use separate models per domain.

**Goal-Conditioned Policy.** To train our goal-conditioned policy, we use the LAVA model (Lynch et al., 2023) architecture, where the text-encoder from CLIP is replaced with a ResNet encoder for the goal image. We train our goal-conditioned policy in each domain using 16 TPUv3 pods for 1 day.

**Baselines.** We compare our approach with a set of multiple baselines. For the LAVA baseline, we follow the exact settings from the LAVA paper (Lynch et al., 2023), where we train the model conditioned on long-horizon goal text and trained the model for 1 day use 16 TPUv3 pods. For our UniPi (Du et al., 2023b) baseline, we followed the same training architecture and approach as the text-conditioned video model in VLP, where the model is trained on long-horizon video elements. For the RT-2 baseline, we follow the architecture and codebase of the original paper (Brohan et al., 2023), where we use the 12B RT2-PaLM-E model. We trained RT-2 using 64 TPUv3 pods for 3 days.

### A.5 PLANNING DETAILS

**Language Table.** To generate video plans, we planned with a horizon of 16, a beam width of 2, a language branching factor of 4, and a video branching factor of 4. To enable fast video generation, we used the DDIM sampler, with a total of 64 timesteps of sampling at the base resolution and 4 timesteps of sampling at the higher resolution samples, with a classifier-free guidance scale of 5 for the base model. We generated 16 videos in parallel at once use a 4 TPU inference pod.

We queried the VLM policy to generate different text actions given an image with a temperature of 0.3. Our VLM heuristic function decoded the number of steps left until task-completion with a temperature of 0.0. We set our heuristic function clipping threshold during planning to be 50 and removed videos if the improvement was larger than 50 after one video rollout. This number was heuristically determined based off generated plans at different thresholds, with the aim of finding the highest threshold such that long-horizon plans still looked physically plausible. Our total planning procedure across a horizon of 16 took approximately 30 minutes.

**7DoF Mobile Manipulator.** In the 7DoF mobile manipulator experiments we made slight tweaks to the approach. For VLM Planning, we used PaLM-E (Driess et al. (2023)) to generate scene captions, and few-shot prompted PaLM (Chowdhery et al. (2022)) to generate plans following the prompts in SayCan (Ahn et al. (2022)). The beam search has a beam width of 3. The video diffusion model is the same as above, except that it has a different resolution for the base model ($64 \times 80$) and super resolution model ($256 \times 320$), and is finetuned on RT-1 (Brohan et al. (2022)) data only. The goal-conditioned policy is using the last frame of the generated video segment only.

**14DoF Bi-Manual manipulation.** We followed the same planning setup as in Language Table. We set our heuristic function clipped threshold during planning to be 15.

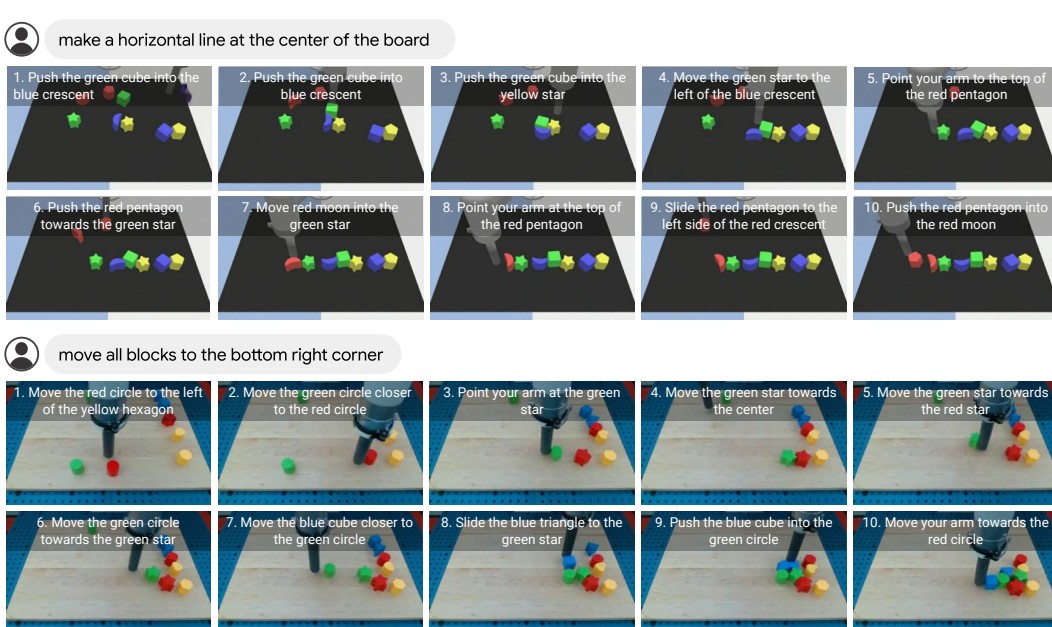

Figure XIV: **Additional Long Horizon Video Plans.** Additional long horizon video plans generated by VLP on both simulated and real images. VLP is *only given* the *initial image* and *language goal*. Language subplans and other image frames are *directly synthesized*.

**Prompt:** Consider a robot table with a red cube at [-0.1, 0.1], red pentagon at [0.1, 0.1], yellow cube at [-0.1, -0.1], yellow heart at [0.2, 0.2], green star at [0.3, 0.3], green pentagon at [-0.3, -0.3], blue cylinder at [0.4, 0.4], blue triangle at [0.5, 0.5]. The robot gripper is at [0, 0]. What is the resultant state of the world after the command, push the red pentagon next to the green star? The robot gripper must physically push blocks. Make sure to simulate the dynamics of gripper and block collision as well as block on block collision. List the state of all blocks and the robot gripper.

**Response:** Simulating the dynamics of the gripper and block collisions, as well as block-on-block collisions, involves series of steps. I'll describe the state of the world after the command "push the red pentagon next to the green star":

Initial state:
•Red cube at [-0.1, 0.1]
•Red pentagon at [0.1, 0.1]
•Yellow cube at [-0.1, -0.1]
•Yellow heart at [0.2, 0.2]
•Green star at [0.3, 0.3]
•Green pentagon at [-0.3, -0.3]
•Blue cylinder at [0.4, 0.4]
•Blue triangle at [0.5, 0.5]
•Robot gripper at [0, 0]

Steps:
1.The robot gripper moves to the red pentagon.
2.The gripper picks up the red pentagon.
3.The gripper moves towards the green star.
4.The gripper releases the red pentagon next to the green star.

Resultant state:
•Red cube at [-0.1, 0.1]
•Yellow cube at [-0.1, -0.1]
•Yellow heart at [0.2, 0.2]
•Green star at [0.3, 0.3]
•Green pentagon at [-0.3, -0.3]
•Blue cylinder at [0.4, 0.4]
•Blue triangle at [0.5, 0.5]
•Robot gripper at [0, 0]
•Red pentagon at [0.45, 0.45] (approximately, based on gripper's release position)

Note: The final position of the red pentagon next to the green star depends on the specific dynamics of the gripper and the release mechanism. The above description assumes a successful execution of the push command without any unexpected behavior or collisions that might alter the final positions.

Figure XV: **Simulating Dynamics with GPT3.5.** LLMs are unable to accurately simulate the visual dynamics of a serialized image representation. In the above example, we provide a simple serialization of an image in Language Table (representing each object as a single point). The LLM is unable to accurately preserve the object in the scene, does not accurately simulate the movement of the gripper and does not simulate the collisions that would happen when the red pentagon move to the green star (collision with the yellow heart).

