# OpenReview forum: "Video Language Planning"
_ICLR.cc/2024/Conference — ICLR 2024 poster_

### Official Review · Reviewer_AYpb · 2023-10-31

**Soundness:** 2 fair
**Presentation:** 3 good
**Contribution:** 2 fair
**Rating:** 6
**Confidence:** 4

**Summary:**

The authors describe an algorithm for video language planning (VLP) that takes as input a long-horizon task instruction and current image observation, and outputs a long video plan that provides detailed multimodal (video and language) specifications that describe how to complete the final task. VLP leverages (i) vision-language models to serve as both policies and value functions, and (ii) text-to-video models as dynamics models. VLP is evaluated on simulated and real-world robotic platforms.

**Strengths:**

1. Overall well-written paper, which is easy to follow
2. Great to see real-world evaluation despite the simplicity of tasks

**Weaknesses:**

1. One of the key contributions of VLP is the use of text-to-video models as dynamics models. But authors experiments and baselines are not convincing about the value of using these models as dynamics models.
* Authors use relatively simple tasks to evaluate VLP. Why not use a text-based LLM or something like this: https://arxiv.org/abs/2106.00188 as a dynamics model instead of the text-to-video model given the simplicity of the evaluation tasks? Would have been great to see such a baseline to really understand the edge that video dynamic models provide.
 * It would have also been great to see VLP being applied to more complex tasks e.g., procedural planning (https://arxiv.org/abs/1907.01172). Generating a walk-through plan for a procedural task is a convincing use case for video dynamics model as compared to say an LLM-based dynamics model.
2. It is also not clear which use-cases will really require VLP as compared to existing LLM-based planning models or more generalized agent architectures such as GATO (https://arxiv.org/abs/2205.06175) or decision transformers. To that end, it would have been useful to see a broader set of baselines in the paper. Some baselines to consider -- RAP: https://arxiv.org/abs/2305.14992 (uses MCTS + LLM dynamics model similar to VLP's tree search+dynamics model), Visual Language Planner: https://arxiv.org/pdf/2304.09179.pdf (learns a multimodal dynamics model and policy by finetuning an LLM), VIMA: https://arxiv.org/pdf/2210.03094.pdf (no dynamics model but multimodal planner, could truly bring out the value of video-based dynamics via such a comparison?)
* Authors also say HiP is the closest to their work in terms of an approach but do not provide a performance comparison in their evaluation. Why? Would love to see it.
3. VLP's inference is slow.  Given that VLP primarily focuses on robotic applications right now, it is unclear whether VLP is suitable for real world deployment. Slow runtime should also be reported as limitations in the main paper (rather than in the appendix).
4. It is unclear if the authors are considering open-sourcing the code and the models. Without that the reproducibility and value of the work for the community reduces. I encourage the authors to discuss open-sourcing plans in their rebuttal.

**Questions:**

- Unclear how long of a video does the video models produce.
- Authors say that they train separate models per domain but it is unclear what is a domain. Is tabletop manipulation in sim and real world considered same or different domain?
- In table. 1, why not ablate VLP without the video dynamics model? Given my concerns in the weakness section, would have loved to see this ablation.

---

> ### Author Response · Authors · 2023-11-20
> **Rebuttal Response**
>
> Thank you for your detailed comments. We respond to each concern below.
>
> **Text LLM for Model Dynamics.** To effectively use a LLM to model dynamics and plan on the precise manipulation settings we consider, the LLM must be able to accurately simulate changes in the visual state subject to each possible natural language action. This requires the first a text-based serialization of the objects in an image as well as resultant simulation based off this serialization. In appendix A.1 (Figure XV), we illustrate that existing LLMs, i.e. GPT-3.5, are unable to accurately simulate the fine dynamics changes of such a serialization.
>
> **Complex Tasks.** In our paper, we illustrate the ability to construct complex video plans across three separate platforms.  in Language Table, we illustrate how our approach can synthesize complex plans to precisely manipulate blocks to form a line. On a  7 DoF mobile manipulator, we illustrate how our approach can synthesize a complex plan to place fruits into a cabinet by first opening the cabinet, putting fruits in it, and finally closing it. Finally, on a 14 DoF dexterous manipulator,  we illustrate how our approach can synthesize  multi-camera video plans of dexterous manipulation involving stacking utensils, cups, and bowls on top of each other. We believe this already represents several axes of difficulty in synthesizing video plans.
>
> **Baseline Comparisons.** Since LLMs are unable to accurately simulate the detailed video dynamics for object manipulation, we cannot apply RAP to our setting. In terms of GATO and VIMA, we already compare VLP against a multimodal transformer, RT-2 which directly predict actions (which both GATO and VIMA can be seen as instances of)  and find that our approach substantially outperforms this baseline. We have added references to each of these papers in the baseline section and clarified our comparisons against them.
>
> **Comparison with HiP.** HiP leverages consensus between a text-to-video and LLM to synthesize a possible language instruction given the current visual observation of the scene. In VLP, we directly learn a VLM to predict text actions to execute given an image observation, which serves as an upper bound on the consensus operation in HiP. Since HiP does not do planning, its performance is upper bounded by the VLP (No Value Function) base which VLP substantially outperforms.  We have clarified this in the main paper.
>
> **Inference Speed.** We have added slow inference as a limitation of our approach. We believe that inference speed can be substantially improved by either distilling the diffusion model, or by using alternative generative decoding methods such as that in Phenalki.
>
> **Code and Models.** We will release an open-source codebase for reproducing the results in VLP by the camera-ready version of the paper. In terms of models – they are trained using a company's internal codebase, so we are unable to directly open-source the models that are trained in the paper. However, we will aim to release pretrained weights for models trained using the released open-source codebase.

---

> > ### Comment · Reviewer_AYpb · 2023-11-22
> > **Post rebuttal**
> >
> > Thank you for the clarifications.
> >
> > I am glad to see the actual numbers on how long does it take for the approach to generate the videos (30 minutes), but it is still not clear to me how long the videos are in the first place.
> >
> > I also appreciated a comparison to a text-based approach for dynamics simulation/state change prediction using GPT in the appendix. However, currently, this example is zero-shot/prompt-based. Given that video dynamics models used by the authors' approach are trained, a true comparison to text-based dynamics approach would have been something that also has access to some training examples -- at the least via in-context learning. I would encourage the authors to think of a fair apples-to-apples comparison between text-based dynamics models and their video dynamics models instead of their current straw-man comparison.
> >
> > Lastly, while considering complex real world tasks as in procedural planning might be out of scope for this work, I would have appreciated atleast a comment on how complex of videos can the approach handle. While I agree that the task of putting fruit in a cabinet is a complex "planning" task,  the vision-based dynamics/videos of this task do not seem complex in comparison to what one would see/expect in real world environments -- think cluttered kitchen counter vs. mono-color table tops with sparse objects in the authors experiments. I'd like to understand if video dynamics models are really the answer to planning in such situations and I don't think the authors provide me that understanding right now.
> >
> > Finally, how would the authors approach compare to models that also learn to generate multimodal tokens e.g. this one: https://arxiv.org/pdf/2304.09179.pdf ?

---

> ### Author Response · Authors · 2023-11-23
> **Author Response**
>
> Thank you for your response, please see our additional clarifications below.
>
> **Duration of Videos.** 30 minutes corresponds to the long horizon video plans shown on the website at https://vlp-iclr2024.github.io/ and corresponds to approximately 15-second videos (for example depicting a robot arranging a set of blocks into a line or moving blocks to a corner of the board).
>
> **Comparisons to Text-Based Dynamics Simulation.** We agree it would be better to compare with a text-based dynamics approach with access to training examples of serialized dynamics predictions. However, our existing training dataset only consists of sequences of images, as opposed to a text serialization, and there is no easy way to convert an image directly into a text representation to train a dynamics model. The most straightforward manner would be directly to tokenize images into multimodal tokens and predict dynamics in this manner and use this for planning (this is actually the first approach we experimented with -- see our response in generating multimodal tokens). If the reviewer believes this baseline is important, we are happy to add such a comparison in the final version of the paper.
>
> **Complexity of Videos.** The video models in our approach are based on those from [1], and as demonstrated in [1] on the Bridge dataset, the video models are able to accurately synthesize dynamics in cluttered kitchen counters as long as there is not significant partial observability (i.e. limited camera movement). However, we found that our approach does not work well when there is substantial camera movement (i.e. when someone is walking down a hallway into another room) as it can not preserve the world state across video frames.
>
> **Generating Multimodal Tokens.** The first approach we explored in our project was actually to directly use the existing PALM-E model to generate image goals using multimodal tokens. Unfortunately, we found that this approach was unable to accurately synthesize image goals in high resolution, perhaps due to the challenge of predicting 256 accurate tokens. Furthermore, this approach could not synthesize video due to context limits as decoding a single video required 16*256=4096 tokens (16x16 tokens per image multiplied by 16 frames).
>
> [1] Learning Universal Policies through Text-Guided Video Generation

---

### Official Review · Reviewer_4qts · 2023-11-01

**Soundness:** 3 good
**Presentation:** 3 good
**Contribution:** 4 excellent
**Rating:** 8
**Confidence:** 3

**Summary:**

This paper addresses the challenge of long-horizon visual planning tasks by utilizing the robust generative capabilities of large vision-language models (VLM) and text-to-video models. The authors introduce the Video Language Planning (VLP) algorithm, which takes both a visual observation and a natural language goal as inputs and subsequently processes a series of instructions, video frames, and low-level controls. The effectiveness of their proposed heuristic value function and tree search procedure is well demonstrated through extensive long-horizon robot manipulation tasks across three hardware platforms.

**Strengths:**

- The long-horizon tasks pose significant challenges and matter in the field of robot manipulation. The proposed VLP algorithm has the potential to significantly improve the success rate of these tasks, as demonstrated in both simulated and real-world experiments in this paper
- Collecting large-scale robot manipulation datasets can be a costly endeavor. VLP, on the other hand, harnesses the impressive generative and generalization capabilities of the latest VLMs and text-to-video models that are pre-trained on Internet-scale data. This approach can serve as valuable inspiration for other researchers looking to tackle challenging robot manipulation tasks without the need for expensive data collection.
- The experimental results regarding the relationship between execution accuracy and planning budget provide valuable insights into the efficiency of the proposed VLP algorithm.

**Weaknesses:**

The primary concern regarding this work is its potential for reproduction and adaptation to, e.g., other hardware platforms and low-cost budgets.
- While the study tests the VLP algorithm on three hardware platforms, they are either relatively simple (i.e., having just one end effector) or self-designed. There is a question as to whether the well-trained VLP model could generalize or adapt quickly to other popular platforms. This limitation may reduce the paper's overall impact.
- While video model inference is computationally expensive, its scalability is also a concern.

**Questions:**

- Table 1 clearly illustrates that the Value Function utilized in VLP significantly enhances result accuracy. However, in many robotic tasks, it is common to have multi-modal policies. For instance, in the "make a horizontal line" task shown in Figure 2, there could be multiple ways to manipulate objects, resulting in various possible remaining steps, especially for images that are far from the final goal. It would be helpful to see whether the PaLM-E model can be fine-tuned to accommodate multi-modal trajectories.
- While the paper properly discusses a few limitations, the limitation of the task horizon is unclear in this paper. In the appendix, the authors provided both VLP and other baselines with 1500 timesteps to complete a task. Does this imply that VLP may not be suitable for handling longer-horizon tasks? For example, a long-distance task, such as moving an object over a long distance (e.g., 10 meters) from one location to another.
- In the introduction, the authors emphasize the potential benefits of VLP when working with incomplete videos that lack corresponding language labels. It would be helpful to see more detailed descriptions and accompanying experimental results to further demonstrate this capability, which are not found in this paper.
- The authors have highlighted the issue of overestimation with the Value Function. It would be interesting to see if out-of-distribution images or goals might also contribute to this problem. If so, could the application of certain offline reinforcement learning algorithms, such as CQL, potentially offer a solution for addressing this issue?
- Minor issues that may need careful proofreading:
  - Sec. 2: a image goal-conditioned -> an
  - as planning submodules Sec. 2.1 -> in Sec. 2.1

---

> ### Author Response · Authors · 2023-11-20
> **Rebuttal Response**
>
> Thank you for your detailed comments and positive evaluation. We respond to each question below.
>
> **Multimodal Value Function.** We would like to clarify that in our experiments with PALM-E, our demonstrations to accomplish long-horizon tasks were already multimodal. For instance, demonstrations for the task of making a line were made by 10 different teleop contractors with very different strategies to arrange a line. To deal with this multimodality, our PALM-E model formulates the task of predicting the number of steps left to finish a task as a language prediction problem. We notice that when sampling from the model, it would thus generate multiple different predictions on the number of steps left to finish a task – corresponding to the different ways to solve this task. To get a single deterministic answer, we directly sampled to 0 temperature decoding of the number of steps left to finish a task, which corresponds to the most likely strategy to finish the task. We have added additional details in Appendix A.4.
>
> **Long Horizon Tasks.** We found that VLP had no difficulties with longer horizon tasks – we artificially set the task length to 1500 so that the evaluation of an episode did not take too long (as a single environment evaluation with 1500 steps already take 1 hour).
>
> **Incomplete Videos Lacking Language Labels.** We would like to clarify that the videos used to train VLP in both Language Table, as well as the 14-DoF dexterous manipulation setting, already lack full language labels. In both settings, videos of the full long-horizon task are collected by teleop workers. However, there is no way then to directly segment videos into corresponding language labels, as there is no clear language label for each consecutive temporal segment of the video. Instead, we then asked a separate group of workers to go through video and select and label any sub-snippets of the video that had coherent language labels. These sub-snippets would often overlap in time in the video and many parts of the video would not have consistent language instructions.  We have clarified this in the main paper as well as in Appendix A.3.
>
> **Overestimation of Value Function.** Yes, we indeed sometimes found that out-of-distribution goals would lead to overestimation of value function . We have added a reference to the CQL paper and methods in offline RL as ways to avoid the exploration of out-of-distribution states.
>
> **Typos.** Thank you, we have fixed the typos you mentioned.

---

> > ### Comment · Reviewer_4qts · 2023-11-23
> >
> > Thank you for your response.
> > I appreciate the additional illustrations provided to enhance the clarity of this paper.
> > It would be great if the fine-tuned 12B PALM-E and the RT2-PaLM-E model could be released.

---

### Official Review · Reviewer_uBLi · 2023-11-05

**Soundness:** 3 good
**Presentation:** 4 excellent
**Contribution:** 3 good
**Rating:** 8
**Confidence:** 3

**Summary:**

The authors propose a general-purpose framework solving for robotics tasks in which planning is done using a video generative model, and then using the sampled video as a target, a low-level controller selects actions to be executed by the robot in order to reproduce the video. To improve the quality of long-term video generation, the authors propose a hierarchical approach in which a list of commands are generated in text-format in addition to the video, and together optimized to minimize an estimate of how close the task is to being completed. The authors demonstrate impressive performance for multiple synthetic and real-world robotics tasks. Importantly, due to the use of internet-scale pre-training for the vision-language model (VLM) and video generative model, the approach generalizes well in a zero-shot setting to novel lighting conditions, objects, and tasks.

**Strengths:**

A great challenge for the machine learning community is to determine how recent advances in generative modeling can be used to advance robotics. I think the method proposed in this paper makes great strides towards this end. Both vision and language foundation models are combined to produce an impressive model-based goal-conditioned planner for robotics. Additionally, I found the paper very well written; I'm not an expert in robotics, but I found the paper easy and interesting to read. The discussion of the limitations at the end is especially insightful, as well as the visuals of video generative model failures in the appendix. The method is stated clearly and its efficacy is backed up with substantial qualitative and quantitative results.

**Weaknesses:**

I think the main weakness of the current submission is that it does not include measurements of runtime for the proposed approach, for instance in Table 3. Planning in video space seems very expensive which could limit how well this approach could be used for robotics tasks that need to be executed quickly. Of course, these runtimes can always be improved, but it would be helpful to see what the runtime is with today's hardware.

Relatedly, I'm curious if the authors have considered optimizing only the text sequence with planning in Algorithm 1, then following with per-clip video generation? It seems much more efficient to me to search through text instead of pixels. Based on Table 3, it seems there are large gains to be had from a more exhaustive search, which I'm guessing is much easier to do if the need to render pixels is eliminated.

**Questions:**

In addition to the two main points mentioned in the weaknesses section, I have a few other minor questions:

- Section 2.2 "Vision-Language Models as Heuristic Functions." - is this model trained with regression, or the tokenized representation of the number of steps with the standard language modeling objective? If it's with the language modeling objective, do the authors choose the mode of the distribution over predicted number of steps when using the heuristic to plan in Algorithm 1?
- Section 2.3 "Replanning" - is there a way to make this re-planning rate dynamic? Ideally, it would be possible to sense when reality has diverged from the video plan in a significant way that is not recoverable.
- A.4 "Video Models" - what's the frequency of these videos, e.g. how many seconds does 16 frames correspond to?

---

> ### Author Response · Authors · 2023-11-20
> **Rebuttal Response**
>
> Thank you for your detailed comments and positive evaluation. We respond to each question below.
>
> **Runtime.** We reported the runtime of our approach in the original submission in Appendix A.2. Our approach takes approximately 30 minutes to synthesize a full long-horizon video and approximately 1 hour to execute a long-horizon task such as making a line of blocks. To make this more visible, we have explicitly added this sampling speed to the limitations sections of the paper. We believe that the runtime can be substantially improved by either distilling the diffusion process or using alternative decoding procedures such as that done in Phenalki.
>
> **Searching with Text.** While it would be desirable to search first for a text plan and then through video prediction, this is difficult because the precise text action to execute next in the text plans depends heavily on the exact continuous state the previous text plan ends on. As an example, consider the text instruction “push the green circle to the left”. If the final goal is to group blocks by color, if the continuous execution of the text instruction pushes it to the left of the green square the next text instruction becomes “push the green circle slightly to the right”, while if its still to the right of the green square the, the text instruction becomes “push the green circle slightly to the left”. Thus to effectively plan, it is important to plan with both text and videos.
>
> **Heuristic Function.** The VLM is trained to predict the number of steps left to execute a task through language prediction on a tokenized representation of the steps left. This allows the VLM to sample a multimodal set of possible number of steps left until task completion. In practice, during search, we sample at 0 temperature from the VLM, corresponding to the mostly likely probability mode of the number of steps left to finish execution of the task. We have clarified this in Appendix A.4
>
> **Replanning.** Yes, replanning can be made dynamic by making replanning occur based off the pixel MSE distance between the current observation and the goal image. We have added a discussion of this in the method section of the paper.
>
> **Video Model Clarification.** The videos are approximately at 15FPS, so 16 frames of video corresponds to one second of video. The long video rollouts shown in the submission cite correspond to between 10-15 video rollouts chained together.

---

### Official Review · Reviewer_MAiq · 2023-11-06

**Soundness:** 2 fair
**Presentation:** 4 excellent
**Contribution:** 3 good
**Rating:** 6
**Confidence:** 4

**Summary:**

The paper proposes a framework for planning in the space of videos and language by leveraging recent text-to-video models (over vision-language models (VLMs) that are trained on static images) for incorporating scene dynamics and object motions. They take as input an initial observation and a long-horizon language instruction and output a plan in the form of video and language specifications, by first prompting a VLM for language actions and rolling out a language-conditioned video model whose outputs are subsequently assessed by the VLM again for favourability towards task progress using a learned heuristic.

**Strengths:**

1) The paper is easy to follow and well-structured with the main contributions listed clearly in the introduction alongwith grounding in related works and building blocks that make up this method (VLM, a video prediction model, and an inverse-dynamics model).
2) The authors present strong model performance compared to 4 baseline methods on object rearrangement tasks, as well as deploy their method on multiple real-robot platforms.

**Weaknesses:**

While the tasks chosen for this paper are claimed to be long-horizon, they are not challenging enough to showcase a big leap in this realm of tasks. For example, when grouping blocks by color, while the task may seem long-horizon given there are large number of blocks on the table to manipulate, there is no dependency between subgoal successes/failures. This makes the task solvable without any need to retain information for long horizons. Hence, I believe the chosen suite of tasks does not evaluate the model adequately for solving long-horizon tasks.

**Questions:**

Are there any limitations induced by video prediction models, which as the authors identified can generate out-of-manifold frames during long-horizon video prediction?

**Details Of Ethics Concerns:**

No concerns.

---

> ### Author Response · Authors · 2023-11-20
> **Rebuttal Response**
>
> Thank you for your detailed comments. We address each of the above issues below.
>
> **Long Horizon Tasks.** In the Language Table domains considered by this work, we believe that there is still some dependency between subgoal successes/failures. In the pair blocks by colors task, a move on one colored block significantly impacts changes the behavior necessary to push another block to pair it. This is further amplified in the make line task, where one incorrect move in one block significantly changes the downstream means necessary in all other blocks. The difficulty of both this pair by color task and the make line task is analyzed in the MAGICAL robust imitation learning benchmark on synthetic 2D shapes [1].  Even in this simplified setting, this benchmark found that existing imitation learning techniques (without planning) get essentially 0 performance on both tasks,  indicating the necessity of reasoning between subgoal successes/failures.
>
> In addition, outside of the Language Table domain, we also evaluate our approach on two other long-horizon tasks. We first illustrate how our approach to solve the task of placing all the fruit on a cabinet on a 7 DoF mobile manipulator, which requires reasoning about both when to open and close cabinets and what objects to put in a cabinet.  We further show how we can synthesize long-horizon video plans for 14-DoF dexterous manipulator to place utensils, cups and bowls together on top of each other. This task again requires a model to be able to reason about which bowls to place on top of each other, as will as cups and utensils.
>
> **Limitations.** We found several limitations with our method.  First, as an image is not a full state representation of the world, our approach cannot deal with occluded objects. Second, our video prediction model does not perfectly preserve the shapes of objects, and they often warp their shape over the video rollout.  Finally, the underlying generation speed of video is very slow (as we run a diffusion process to generate each video snippet) preventing the approach from be used for realtime control. We have added additional limitations in the limitations section of the paper in Section 5.
>
> [1] Toyer et al. The MAGICAL Benchmark for Robust Imitation

---

### Meta-Review · Area_Chair_Phy6 · 2023-12-10

**Metareview:**

This paper has received overall positive ratings (8, 8, 6, 6). The reviewers acknowledged that 1) the paper tackles an important problem in a timely manner, 2) the proposed approach is sufficiently novel, 3) experiments are convincing, and 4) the paper is well-written. Given the merits of the paper, we recommend acceptance.

**Justification For Why Not Higher Score:**

There were some concerns about high compute budget and somewhat simplistic task scenario.

**Justification For Why Not Lower Score:**

Despite the weaknesses, the paper's merit on the timely topic and convincing experimental results (despite in a relatively simple setting) is encouraging and should be of interest to the ICLR community.

---

### Decision · Program_Chairs · 2024-01-16

Accept (poster)